# Adaptive Image Quality Assessment via Teaching Large Multimodal Model to Compare

**Hanwei Zhu**[*1]  **Haoning Wu**[*2]  **Yixuan Li**[1]  **Zicheng Zhang**[3]  **Baoliang Chen**[4]
**Lingyu Zhu**[1]  **Yuming Fang**[5]  **Guangtao Zhai**[3]  **Weisi Lin**[2]  **Shiqi Wang**[†1,6]

[1] City University of Hong Kong
[2] Nanyang Technological University
[3] Shanghai Jiao Tong University
[4] South China Normal University
[5] Jiangxi University of Finance and Economics
[6] Shenzhen Research Institute, City University of Hong Kong

https://compare2score.github.io/

## Abstract

While recent advancements in large multimodal models (LMMs) have significantly improved their abilities in image quality assessment (IQA) relying on *absolute* quality rating, how to transfer reliable *relative* quality comparison outputs to continuous perceptual quality scores remains largely unexplored. To address this gap, we introduce **Compare2Score**—an all-around LMM-based no-reference IQA (NR-IQA) model, which is capable of producing qualitatively comparative responses and effectively translating these discrete comparative levels into a continuous quality score. Specifically, *during training*, we present to generate scaled-up comparative instructions by comparing images from the same IQA dataset, allowing for more flexible integration of diverse IQA datasets. Utilizing the established large-scale training corpus, we develop a human-like visual quality comparator. *During inference*, moving beyond binary choices, we propose a soft comparison method that calculates the likelihood of the test image being preferred over multiple predefined anchor images. The quality score is further optimized by maximum a posteriori estimation with the resulting probability matrix. Extensive experiments on nine IQA datasets validate that the **Compare2Score** effectively bridges **text-defined comparative levels** during training with converted **single image quality score** for inference, surpassing state-of-the-art IQA models across diverse scenarios. Moreover, we verify that the probability-matrix-based inference conversion not only improves the rating accuracy of **Compare2Score** but also zero-shot general-purpose LMMs, suggesting its intrinsic effectiveness.

## 1 Introduction

Image quality assessment (IQA) models aim to establish a quantitative mapping between digital visual images and human subjective evaluations, playing an indispensable role across various image processing and computer vision tasks [1]. No-reference IQA (NR-IQA) [2, 3, 4, 5], which evaluate images without a reference, are particularly valuable for real-world applications. Recently, NR-IQA has experienced profound improvement through advanced deep neural networks (DNNs) [6, 7, 8, 9,

---

[*]Equal contribution
[†]Corresponding author: shiqwang@cityu.edu.hk

38th Conference on Neural Information Processing Systems (NeurIPS 2024).

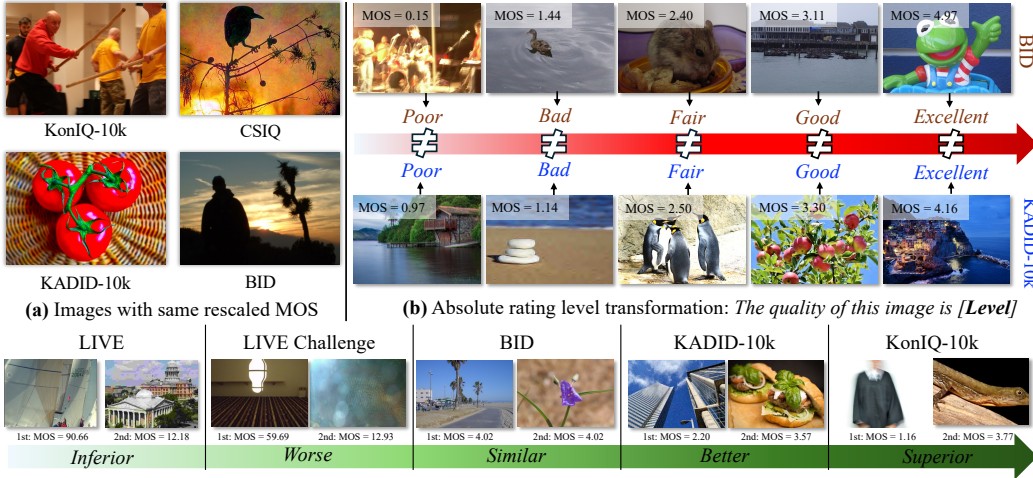

**(a)** Images with same rescaled MOS

**(b)** Absolute rating level transformation: *The quality of this image is [**Level**]*

**(c)** Relative comparison level transformation: *The quality of the 2nd image is [**Level**] to/than the 1st image*

Figure 1: Illustrations of the motivation of this work. **(a)** Images with identical rescaled MOS from various IQA datasets exhibit significant variations in perceptual quality. **(b)** Images that cluster at the same rating level from different IQA datasets display mismatches due to differing subjective testing methodologies. **(c)** By comparing MOSs within the same dataset, it facilitates the flexible combination of multiple IQA datasets.

10]. However, a primary challenge with current models [8, 11, 12] lies in their limited *cross-distortion* generalization capability since the training and testing data contain significant distribution shift.

To improve the generalization capability of the NR-IQA, lots of advanced training techniques have been adopted, such as meta learning [13], domain adaptation [14], test-time adaption [15], and hard example mining [16]. More capable foundation models, such as CLIP [10] and large multimodal models (LMMs) [12], are also proven to be effective in improving generalization ability. Despite these advancements, the gains from such techniques remain constrained due to the persistent **data challenge** inherent in IQA. Alternatively, expanding the IQA training datasets—both in terms of the number of images and the diversity of distortions—emerges as a scalable strategy to augment model robustness [17]. This data scaling law has also been recognized as one of the key factors in building effective LMMs [18, 19, 20, 21, 22, 23]. As such, addressing how to effectively combine existing IQA datasets to meet the extensive data requirements of training LMMs is highly desirable.

As an early attempt of LMMs on IQA, Q-Align [12] proposed to combine different IQA datasets with *absolute* quality ratings. While absolute ratings are widely used for collecting human opinions on different IQA datasets, it is non-trivial to directly fuse them for LMM training. This difficulty arises because each dataset has **different perceptual scales** owing to varying subjective testing methodologies. As shown in Fig. 1 (a), images with identical mean option score (MOS) (2.3, all rescaled to range $[0, 5]$) from four datasets [24, 25, 26, 7] differ significantly in perceptual quality. As a result, despite clustering at the same rating level, these images from different datasets display distinctly different visual qualities (see Fig. 1 (b)). Therefore, simply scaling up the training data by mixing existing IQA datasets with rescaled MOS is fundamentally flawed. Instead, the *relative* quality ranking (*e.g.*, paired comparison) offers intrinsic simplicity and reliability over absolute quality rating [27]. As shown in Fig. 1 (c), it is comparable to the images from the same IQA datasets as their MOSs originate from the same subjective user study, facilitating a more flexible combination of various IQA datasets. However, the key limitation of the paired comparison method is its impracticality in deriving individual image quality scores from $\binom{M}{2}$ comparisons when $M$ is large. Furthermore, the lack of effective and efficient methods to convert *relative* comparisons into quantitative *absolute* ratings makes current comparison-based approaches difficult to apply to real-world scenarios.

To tackle these challenges, this paper leverages the flexibility and reliability of relative quality comparisons to introduce **Compare2Score**—an all-around LMM-based NR-IQA model, which is

designed to generate human-like qualitative comparisons and compute effective quality scores. Before delving into detail, we clearly highlight our main contributions as follows.

- **[A repurposed training dataset.]** We introduce a tailored approach to generate comparative instructions by comparing MOSs within each IQA dataset. This method categorizes image pairs into distinct comparative levels (`inferior`, `worse`, `similar`, `better`, `superior`) using the empirical rule, facilitating the flexible integration of diverse IQA datasets. This specific implementation effectively addresses the challenges posed by differing subjective testing methodologies and perceptual scales. It produces a comprehensive training dataset that enables the LMM to handle various distortion scenarios, resulting in a human-like **visual quality comparator**.

- **[An inference conversion strategy.]** We develop an **adaptive soft comparison scheme** that efficiently translates discrete comparative levels into continuous quality scores. Unlike traditional two-alternative forced choice (2AFC) methods, our approach calculates the likelihood that an input image is preferred over multiple anchor images. This probability is derived from a weighted summation of the softmax-transformed log probabilities across five comparative levels. Subsequently, the quality score of the input image is calculated through maximum a posteriori (MAP) estimation based on the resulting probability matrix.

- **[A state-of-the-art framework.]** We conduct extensive experiments to validate the effectiveness of teaching the relative quality ranking knowledge to LMM. The proposed model, namely **Compare2Score**, consistently outperforms state-of-the-art NR-IQA models on both synthetic and realistic distortions and shows enhanced generalization capability across different cross-distortion scenarios. Furthermore, we demonstrate that the probability matrix-based inference conversion significantly enhances the rating accuracy of **Compare2Score** and extends these improvements to zero-shot general-purpose LMMs.

## 2 Related Work

### 2.1 NR-IQA Models

**Regressing for NR-IQA** Traditional learning-to-regress NR-IQA models [28, 29, 30, 31] build effective quality-aware feature extractors rooted in theoretical principles, which are then mapped to quality scores through well-trained IQA regressors. In contrast, deep-learning-based IQA models [6, 9, 32, 33] exploit large volumes of IQA data to simultaneously refine DNNs for both feature extraction and quality regression. By using advanced training techniques, such as domain adaption [5], meta-learning [13], multi-task learning [34, 10], and contrastive learning [32], NR-IQA models show high correlation with the HVS. However, these models show limited cross-distortion ability. Additionally, these models typically produce quantitative quality scores, creating a significant gap from subjective user studies that prefer learning and assigning text-defined quality levels [35].

**Ranking for NR-IQA** Beyond learning-to-regress schemes, many models address IQA through a relative quality ranking setting [36, 37, 38, 17, 10]. Liu *et al.* [37] synthesized a large IQA dataset labeled with distortion types and levels to train a Siamese network for precise image quality ranking. Zhang *et al.* [17] incorporated probabilistic quality preference for image pairs from diverse datasets to address inter-dataset incomparable concerns. LIQE utilized a pairwise learning-to-rank training strategy with both visual and textual inputs [10]. Such ranking-based models mitigate the vulnerability towards task-agnostic information of regressing-based models, enabling more robust capabilities for different distortion scenarios [17, 10]. Nevertheless, the application of the learning-to-rank scheme for LMM is largely under-explored. As such, we present to leverage relative comparisons to develop an LMM-based NR-IQA model that produces qualitative comparison outcomes and translates the discreet comparative levels into continuous quality scores effectively.

### 2.2 LMMs for IQA

Recently, many works have explored the capabilities of LMMs on IQA, covering both benchmarking [39, 40, 41, 42] and refining [39, 12, 43, 44, 45, 46]. Wu *et al.* laid the groundwork by examining and instructing of LMMs in low-level vision tasks, through the development of Q-Bench [47] and Q-Instruct [39], respectively. Wu *et al.* analyzed LMM's performance under various standardized

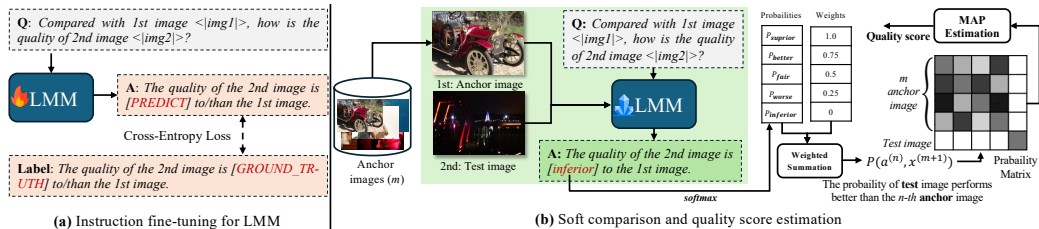

**(a)** Instruction fine-tuning for LMM

**(b)** Soft comparison and quality score estimation

Figure 2: Training and inference phash of **Compare2Score**. **(a)** The LMM is fine-tuned with instruction-response pairs generated by comparing the MOSs from the same IQA dataset, allowing for a more flexible combination of various IQA datasets. **(b)** The trained visual quality comparator (*i.e.*, LMM) is utilized to compute the likelihood of a test image being preferred over the anchor images, and then the quality score is derived using MAP estimation.

prompting settings [42], validating the effectiveness of chain-of-thought prompting in IQA tasks. Co-Instruct [46] extended the low-level visual capability of the LMM to meet the requirement of multiple image inputs. Despite demonstrating success, the qualitative outputs of the above LMMs are hard to transfer to a quantitative score, which often plays an important role in computer vision tasks [48]. Q-Align [12] made the first attempt by transferring the qualitative rating levels to the perceptual quality scores. However, as shown in Figs. 1 (a) & (b), it is impractical to expand the training dataset by simply mixing multiple IQA datasets with rescaled MOS. Therefore, we introduce a novel relative ranking strategy that allows for the seamless integration of existing IQA datasets into an expanded training set, which is then utilized to train an LMM-based visual quality comparator.

## 3 The Compare2Score Framework

In this section, we first describe the preliminaries of paired comparison for humans and LMMs, respectively. Subsequently, we introduce our methodological framework, which includes generating the training data with quantitatively comparative instructions (Sec. 3.2), the LMM-based visual quality comparator (Sec. 3.3), and the soft comparison method for quality score derivation (Sec. 3.4). Fig. 2 shows the training and testing diagrams of **Compare2Score** framework.

### 3.1 Preliminaries

**Paired Comparison for Humans.** The paired comparison methodology in subjective testing involves three principal steps: the combination of images, the collection of human judgments, and the aggregation of these judgments into quality scores. In particular, given a set of images $\mathcal{X} = \{(x^{(i)}, q^{(i)})\}_{i=1}^M$ where $x^{(i)} \in \mathbb{R}^N$ is the image and $q^{(i)} \in \mathbb{R}$ represents the ground-truth quality score, the methodology requires comparing a total of $\binom{M}{2}$ pairs. We assume that a higher $q^{(i)}$ indicates better perceptual quality. The outcomes of these comparisons are recorded in a count matrix $C \in \mathbb{R}^{M \times M}$, where each entry records the number of times one image is preferred over another. The global ranking scores $\hat{q} = \{\hat{q}^{(i)}\}_{i=1}^M$ can be computed by MAP estimation [49]:

$$\arg\max_{\hat{q}} \mathcal{L}(\hat{q}|C) + \log p(\hat{q}), \text{ s.t. } \sum_i \hat{q}^{(i)} = 0, \tag{1}$$

where $\mathcal{L}(\cdot)$ denotes the log-likelihood function and $p(\hat{q})$ is a prior on the scale values. Although effective [27], the key limitation of paired comparison is the exponential growth in the number of comparisons, which becomes labor-intensive and costly for large $M$. Moreover, once the experiment concludes, incorporating new test images for quality score inference becomes almost infeasible.

**Paired Comparison for LMMs.** Inspired by the efficacy and reliability of paired comparison experiments with humans, we explore the feasibility of adapting this approach for LMMs. Accordingly, we adopt a similar pipeline to that used in subjective testing. The framework for training LMMs and predicting quality scores also involves three core steps: constructing instruction-response pairs, fine-tuning the LMM with such pairs, and inferring quality scores. To increase the efficiency and feasibility of the model, we propose an adaptive soft comparison approach by computing the probability of the test image $x^{(i)}$ being preferred over $m$ representative anchor images. The probability is

calculated by a weighted summation of the softmax of the log probabilities across five comparative levels. The outcome of the LMM is a probability matrix, $P \in \mathbb{R}^{(m+1) \times (m+1)}$ where $m \ll M$. The quality score of the input image is then computed by MAP estimation with the same optimization problem as Eqn. (1).

## 3.2 Training Dataset: Comparative Instruction-Response Pairs

To facilitate the reasonability of mixing different IQA datasets, we present to compare the visual quality of pairs of images within each IQA dataset. It allows a seamless combination of $K$ established IQA databases for generating large-scale repurposed training dataset (*i.e.*, instruction-response pairs). This process involves translating MOSs into discrete comparative levels. Utilizing the empirical rule [50], we define five levels of comparison: `inferior`, `worse`, `similar`, `better`, `superior`. The format of the instruction-response pairs is specified as follows:

USER: Compared with the first image <img1>, how is the quality of the second image <img2>?
ASSISTANT: The quality of the second image is [Level] to/than the first image.

Specifically, we randomly sampling $n_k$ image pairs $\{(x_k^{(i)}, x_k^{(j)})\}_{i,j=1}^{n_k}$ from each database. For each pair $\{(x^{(i)}, x^{(j)})\}$, relative quality rankings are inferred based on MOS and its standard deviation. We assume the perceptual quality of each image $x^{(i)}$ as a Gaussian distribution, characterized by mean $q^{(i)}$ and standard deviation $\sigma^{(i)}$, derived from subjective testing.

Assuming independence in quality variability between images, the quality differential also follows a Gaussian distribution with mean $q^{(ij)} = q^{(i)} - q^{(j)}$ and standard deviation $\sigma^{(ij)} = \sqrt{(\sigma^{(i)})^2 + (\sigma^{(j)})^2}$. This methodology, summarized in Eqn. (2), introduces sig-

$$\text{Level} = \begin{cases} \text{inferior} & \text{if } q^{(ij)} > 2\sigma^{(ij)}, \\ \text{worse} & \text{if } \sigma^{(ij)} < q^{(ij)} \leq 2\sigma^{(ij)}, \\ \text{similar} & \text{if } -\sigma^{(ij)} < q^{(ij)} \leq \sigma^{(ij)}, \\ \text{better} & \text{if } -2\sigma^{(ij)} < q^{(ij)} \leq -\sigma^{(ij)}, \\ \text{superior} & \text{if } q^{(ij)} < -2\sigma^{(ij)} \end{cases} \quad (2)$$

nificance thresholds at $\pm\sigma^{(ij)}$ and $\pm2\sigma^{(ij)}$, effectively categorizing the quality differences into meaningful comparative levels. These thresholds function similarly to confidence intervals in statistical hypothesis testing, establishing a robust framework for accurately identifying significant perceptual differences between images.

## 3.3 Structure: Multi-image LMM as Visual Quality Comparator

The visual quality comparator forms a central element within the **Compare2Score** framework and is tasked with predicting qualitative judgments for pairs of images. As shown in Fig. 3, the architecture incorporates the advanced mPLUG-Owl2 model [20], which comprises the image encoder ($f_\psi$), the image abstractor ($f_\delta$), and the large language model (LLM) decoder ($g_\phi$). The process begins with the image encoder transforming each image into a visual embedding. This embedding is then dimensionally reduced by the image abstractor to facilitate the handling of multiple images, expressed as $z^{(i)} = f_\delta(f_\psi(x^{(i)}))$, where $z^{(i)} \in \mathbb{R}^U$ with $U = 65$, significantly less than LLaMA-2's maximum context length of $2,048$ [51]. These compact visual embeddings are combined with textual embeddings $t \in \mathbb{R}^V$ from the text tokenizer and projected into a shared semantic space. The LLM decoder takes the aligned features and interleaved them to produce the qualitative output, formalized as $\texttt{Output} = g_\phi(< z^{(i)}, t >)$, where $< \cdot, \cdot >$ represents the feature alignment.

## 3.4 Inference Conversion: Adaptive Soft Comparison

**Soft Comparison Methodology.** After training, the response is determined by selecting the token with the highest probability from the LLM decoder. This conventional method, while straightforward, may not fully exploit the nuanced capabilities of LMMs, as it relies solely on the most probable outcome and disregards other informative probabilities. To overcome this limitation, we propose a soft comparison method that integrates the logits of all five comparative tokens $\mathcal{T} = \{t_i|_{i=1}^5\} = \{\texttt{inferior}, \texttt{worse}, \texttt{similar}, \texttt{better}, \texttt{superior}\}$. The probability of each token is achieved by the softmax function, expressed as $p_{e^{t_i}} = e^{t_i} / \sum_{j=1}^5 e^{t_j}$, where $p_{e^{t_i}}$ indicate the probability of $i$-th token. Moreover, to enhance the efficiency and feasibility of the model, we do not compare the test

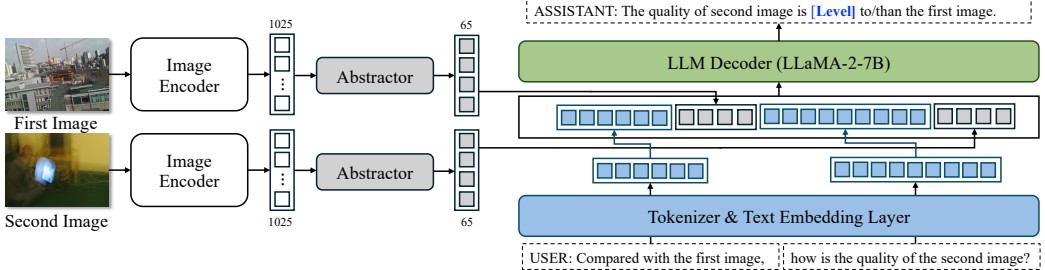

Figure 3: Architecture of the proposed **Compare2Score**. Images are initially processed by an image encoder, followed by token reduction through an abstractor module. The aligned textual and visual embedding are interleaved and processed by the large language model (LLM) decoder to generate precise qualitative comparative levels for paired comparisons.

image against every image in the dataset. Instead, we identify a smaller set of anchor images from the training set, denoted as $\mathcal{A} = \{a^{(n)}\}_{n=1}^{m}$, where $a^{(n)}$ represents the $n$-th anchor image. As a result, the probability of the test image being preferred over the anchor images is computed by the weighted summation:

$$P_{e^{t_i}}(a^{(n)}, x^{(m+1)}) = \sum_{i=1}^{5} w_i p_{e^{t_i}}(a^{(n)}, x^{(m+1)}), \; n = 1 \ldots m, \tag{3}$$

where $w_i$ are predefined weights $\{w_i|_{i=1}^{5}\} = \{0, 0.25, 0.5, 0.75, 1\}$, facilitating nuanced interpretation and use of the comparative levels.

**Anchor Image Selection.** We initially partition the IQA dataset into $\alpha$ quality intervals, represented as $\mathcal{X} = \bigcup_{i=1}^{\alpha} \mathcal{X}(i)$. Our objective is to identify $\beta$ representative images from each quality interval, characterized by minimal variability in their MOS scores, enhancing the consistency of our experimental dataset. Images with high variability in human ratings are deemed less suitable for evaluating the performance of LMMs due to the potential introduction of noise and biases. For each interval $\mathcal{X}(i)$, we aim to select a subset $\mathcal{A}(i) \subseteq \mathcal{X}(i)$, where the size of $\mathcal{A}(i)$ is $\beta$. This selection criterion is formalized through the following optimization problem:

$$\mathcal{A}(i) = \underset{\mathcal{A} \subseteq \mathcal{X}(i), |\mathcal{A}| = \beta}{\arg \min} \sum_{x \in \mathcal{A}} \sigma(x)^2, \tag{4}$$

where $\sigma(x)^2$ denotes the variance of the MOS score for image $x$, serving as a quantitative measure of rating consistency. As such, the full set of anchor images can be achieved by $\mathcal{A} = \bigcup_{i=1}^{\alpha} \mathcal{A}(i)$

**Probability Matrix Construction.** Based on the selected anchor images and visual quality comparator, we first construct probability matrix $P_a \in \mathbb{R}^{m \times m}$ for the anchor images as follows:

$$P_a = \begin{bmatrix} P(a^{(1)}, a^{(1)}) & P(a^{(1)}, a^{(2)}) & \cdots & P(a^{(1)}, a^{(m)}) \\ P(a^{(2)}, a^{(1)}) & P(a^{(2)}, a^{(2)}) & \cdots & P(a^{(2)}, a^{(m)}) \\ \vdots & \vdots & \vdots & \vdots \\ P(a^{(m)}, a^{(1)}) & P(a^{(m)}, a^{(2)}) & \cdots & P(a^{(m)}, a^{(m)}) \end{bmatrix} \tag{5}$$

Notably, each element $P(a^{(i)}, a^{(j)}) = 1 - P(a^{(j)}, a^{(i)})$ and $P(a^{(i)}, a^{(i)}) = 0.5$. Then, the test image $x^{(m+1)}$ is compared with all anchor images. We use $b = \left[ P(a^{(1)}, x^{(m+1)}), P(a^{(2)}, x^{(m+1)}), \ldots, P(a^{(m)}, x^{(m+1)}) \right]$ to denote the resultant vector. Therefore, we finally form the complete probability matrix $P \in \mathbb{R}^{(m+1) \times (m+1)}$ for the anchor and test images as $P = \begin{bmatrix} P_a & b \\ (1-b) & 0.5 \end{bmatrix}$.

**Quality Score Estimation.** Once obtaining the probability matrix, we compute the quality scores using MAP estimation under Thurstone's Case V model [52]. It is expressed as a convex optimization

problem [49]:

$$\arg\max_{\hat{q}} P_{ij} \log(\Phi(\hat{q}^{(i)} - \hat{q}^{(j)})) - \sum_i \frac{\hat{q}^{(i)}}{2}, \text{ s.t. } \sum_i \hat{q}^{(i)} = 0, \tag{6}$$

where $\Phi(\cdot)$ is the standard Normal cumulative distribution function, and $\hat{q}^{(m+1)}$ represents the quality score of the test image.

## 4 Experiments

### 4.1 Experimental Setups

**IQA Datasets.** We conduct comprehensive experiments across six standard IQA datasets. These datasets are categorized based on the type of distortions they contain: synthetic distortions are featured in LIVE [53], CSIQ [24], and KADID-10k [26]; realistic distortions are present in BID [25], LIVE Challenge (denoted as CLIVE) [54], and KonIQ-10k [7]. More details regarding these IQA datasets can be found in the Appendix A.2. For our experiments, we utilize the ten splits provided by LIQE[3], allocating 70% of images from each dataset for training, 10% for validation, and the remaining 20% for testing. We combine the training and validation sets in our experiments since **Compare2Score** are evaluated at the last optimization iteration without any fine-tuning of the model parameters [12, 46]. For datasets with synthetic distortions, it strictly maintains content independence by splitting datasets using reference images [10]. The median of the Spearman's rank correlation coeff icient (SRCC) and Pearson linear correlation coefficient (PLCC) across the ten splits are reported in the tables.

**Implementation Details.** **Compare2Score** utilizes the advanced mPLUG-Owl2 model [20] for its architecture, leveraging a pre-trained CLIP-ViT-L14 as the vision encoder [55] and LLaMA2-7B [51] as the LLM decoder. To train the model, we generate $180,000$ image pairs and optimize the whole architecture with the GPT loss [56], which computes cross-entropy between the predicted logits and ground-truth labels. Training is conducted with a batch size of 64 across all datasets, a fixed learning rate of $2 \times 10^{-5}$, and spans two epochs. This process requires seven NVIDIA A40 GPUs to meet the computational load. During inference, a single NVIDIA RTX3090 GPU is sufficient for executing the soft comparison (Sec. 3.4). Furthermore, to obtain the anchor images, we divide the training set of the KonIQ-10k into five ($\alpha = 5$) quality intervals based on their MOSs [57], from which we select one ($\beta = 1$) representative anchor image per interval using Eqn. (4).

**Baselines.** We compare the performance of the proposed **Compare2Score** with the following state-of-the-art methods, which include (1) three opinion-unaware NR-IQA models: NIQE [2], ILNIQE [3], and Ma19 [58]; (2) six learning-to-regress NR-IQA models: PaQ2PiQ [6], KonCept [7], MUSIQ [59], DBCNN [4], HyperIQA [8], and TreS [11]; (3) two learning-to-rank NR-IQA models: UNIQUE [17] and LIQE [10]; (4) one LMM-based NR-IQA model: Q-Align [12]. All methods are compared with the same testing sets across ten splits. The UNIQUE, LIQE, and Q-Align are jointly trained on the above six datasets, respectively. The remaining methods are separately trained on each individual dataset if necessary.

### 4.2 Main Results

**Performance under Intra-Dataset Setting.** Table 1 shows the results of media SRCC and PLCC across ten sessions. It is clear that **Compare2Score** outperforms the competing methods on both synthetic and realistic distortions, demonstrating the reliability of the paired comparison strategy can be smoothly extended to the LMM. While Q-Align [12] is another LMM-based model, it presents inferior performance on the synthetically distorted datasets [53, 24, 26]. The main reason may be the perceptual scale ambiguity across different IQA datasets. The pairwise learning-to-rank approach employed by UNIQUE [17] and LIQE [10] achieves competitive performance against models [4, 8, 11] trained individually, which further validates the effectiveness of using relative ranking information to mix different IQA datasets. Additionally, the opinion-unaware models exhibit subpar performance on the realistic distortions, suffering the potential overfitting issue to the traditional synthetic distortions [53, 24]. Furthermore, though the learning-to-regress models are

---

[3]https://github.com/zwx8981/LIQE/tree/main/IQA_Database

Table 1: Performance comparison in terms of median SRCC and PLCC on six IQA datasets. The methods are jointly trained with a mixture of six datasets are represented in italics.

| Method | LIVE [53] SRCC | PLCC | CSIQ [24] SRCC | PLCC | KADID-10k [26] SRCC | PLCC | BID [25] SRCC | PLCC | CLIVE [54] SRCC | PLCC | KonIQ-10k [7] SRCC | PLCC |
|---|---|---|---|---|---|---|---|---|---|---|---|---|
| NIQE [2] | 0.908 | 0.904 | 0.631 | 0.719 | 0.389 | 0.442 | 0.573 | 0.618 | 0.446 | 0.507 | 0.415 | 0.438 |
| ILNIQE [3] | 0.887 | 0.894 | 0.808 | 0.851 | 0.565 | 0.611 | 0.548 | 0.494 | 0.469 | 0.518 | 0.509 | 0.534 |
| Ma19 [58] | 0.922 | 0.923 | 0.926 | 0.929 | 0.465 | 0.501 | 0.373 | 0.399 | 0.336 | 0.405 | 0.360 | 0.398 |
| PaQ2PiQ [6] | 0.544 | 0.558 | 0.697 | 0.766 | 0.403 | 0.448 | 0.719 | 0.700 | 0.732 | 0.755 | 0.722 | 0.716 |
| KonCept [7] | 0.673 | 0.619 | 0.631 | 0.645 | 0.503 | 0.515 | 0.816 | 0.825 | 0.778 | 0.799 | 0.911 | 0.924 |
| MUSIQ [59] | 0.837 | 0.818 | 0.697 | 0.766 | 0.572 | 0.584 | 0.744 | 0.774 | 0.785 | 0.828 | 0.915 | **0.937** |
| DBCNN [4] | 0.963 | 0.966 | **0.940** | **0.954** | 0.878 | 0.878 | 0.864 | 0.883 | 0.835 | 0.854 | 0.864 | 0.868 |
| HyperIQA [8] | 0.966 | **0.968** | 0.934 | 0.946 | 0.872 | 0.869 | 0.848 | 0.868 | 0.855 | 0.878 | 0.900 | 0.915 |
| TreS [11] | 0.965 | 0.963 | 0.902 | 0.923 | 0.881 | 0.879 | 0.853 | 0.871 | 0.846 | 0.877 | 0.907 | 0.924 |
| *UNIQUE* [17] | 0.961 | 0.952 | 0.902 | 0.921 | 0.884 | 0.885 | 0.852 | 0.875 | 0.854 | 0.884 | 0.895 | 0.900 |
| *LIQE* [10] | **0.970** | 0.951 | 0.936 | 0.939 | **0.930** | **0.931** | 0.875 | 0.900 | 0.904 | 0.910 | 0.919 | 0.908 |
| *Q-Align* [12] | 0.913 | 0.919 | 0.915 | 0.936 | 0.869 | 0.927 | **0.904** | **0.920** | **0.931** | 0.921 | **0.935** | 0.934 |
| *Compare2Score* | **0.972** | **0.969** | **0.950** | **0.943** | **0.952** | **0.939** | **0.919** | **0.939** | **0.914** | **0.928** | **0.931** | **0.939** |

Table 2: SRCC results on the three IQA datasets under the cross-dataset setup. The methods are jointly trained with a mixture of six datasets are represented in italics.

| Method | TID2013 [24] | SPAQ [60] | AGIQA-3K [61] |
|---|---|---|---|
| NIQE [2] | 0.314 | 0.578 | 0.562 |
| PaQ2PiQ [6] | 0.423 | 0.823 | 0.502 |
| MUSIQ [59] | 0.584 | 0.853 | 0.629 |
| DBCNN [4] | 0.686 | 0.412 | 0.654 |
| HyperIQA [8] | - | - | 0.629 |
| Tres [62] | - | - | 0.646 |
| *UNIQUE* [17] | 0.768 | 0.838 | 0.666 |
| *LIQE* [10] | 0.811 | 0.881 | 0.721 |
| *Q-Align* [12] | 0.801 | 0.813 | 0.725 |
| *Compare2Score* | **0.823** | **0.906** | **0.730** |

Table 3: SRCC results of probability matrix and count matrix on four IQA datasets. Prob. stands for probability.

| Method | Matrix | LIVE [53] | CSIQ [24] | BID [25] | CLIVE [54] |
|---|---|---|---|---|---|
| IDEFICS2 [19] | Count | 0.354 | 0.208 | 0.198 | 0.292 |
| | Prob. | **0.465** | **0.567** | **0.389** | **0.436** |
| LLaVA-1.5 [21] | Count | 0.214 | 0.148 | 0.122 | 0.015 |
| | Prob. | **0.386** | **0.555** | **0.361** | **0.292** |
| mPLUG-Owl2 [20] | Count | 0.408 | 0.013 | 0.217 | 0.221 |
| | Prob. | **0.449** | **0.129** | **0.551** | **0.355** |
| XComposer-VL-2 [22] | Count | 0.199 | 0.145 | 0.206 | 0.332 |
| | Prob. | **0.323** | **0.301** | **0.598** | **0.455** |
| Co-Instruct [46] | Count | 0.582 | 0.569 | 0.820 | 0.694 |
| | Prob. | **0.822** | **0.768** | **0.866** | **0.768** |
| **Compare2Score** | Count | 0.888 | 0.875 | 0.778 | 0.816 |
| | Prob. | **0.974** | **0.942** | **0.921** | **0.934** |

able to achieve promising performance on individual datasets with unique parameters, each dataset requires a unique set of parameters that hinder the practicality of such models in the real world.

**Performance under Cross-Dataset Setting.** To assess the generalization capability of **Compare2Score** against competitive NR-IQA models, we conduct the cross-distortion experiments with three challenging unseen IQA datasets: TID2013 [63], SPAQ [60], and AGIQA-3K [61]. In particular, TID2013 contains 24 distortion types, most of which are different from distortions in the training datasets. SPAQ consists of 11, 125 images captured by 66 smartphones, undergoing abundant realistic distortions. The images in AGIQA-3K are generated by six advanced text-to-image generative models, which cast significant challenges to the NR-IQA models. The results are summarized in Table. 2, from which we can observe that **Compare2Score** demonstrates the strongest generalization capability across synthetic, realistic, and generative distortions. We believe the robustness of the proposed model benefits from 1) the high capacity of the LMM-based model, 2) the proposed soft comparison mechanism, and 3) the joint training on multiple datasets.

**Performance of Probability Matrix and Count Matrix.** In order to demonstrate the efficacy of our proposed soft comparison method, we conducted an evaluation comparing the newly designed probability matrix against the traditional count matrix [49] with five state-of-the-art LMMs, including IDEFICS2 [19], LLaVA-1.5 [21], mPLUG-Owl2 [20], XComposer-VL-2 [22], and Co-Instruct [46]. Detailed information about these models is available in the Appendix A.3. The comparative results are detailed in Table 3. The results reveal that our probability matrix not only enhances the performance of **Compare2Score** but also significantly zero-shot the IQA performance across five open-source LMMs [19, 21, 20, 22, 46]. This consistent outperformance highlights the robustness and utility of the soft comparison approach in diverse IQA contexts.

Table 4: Performance comparison in terms of prediction accuracy on six IQA datasets. The best results are highlighted in boldface.

| Method | LIVE [53] | CSIQ [24] | KADID-10k [26] | BID [25] | CLIVE [54] | KonIQ-10k [7] |
|---|---|---|---|---|---|---|
| IDEFICS2 [19] | 0.453 | 0.546 | 0.521 | 0.566 | 0.407 | 0.687 |
| LLaVA-1.5 [21] | 0.170 | 0.544 | 0.600 | 0.579 | 0.074 | 0.455 |
| mPLUG-Owl2 [20] | 0.484 | 0.394 | 0.302 | 0.613 | 0.407 | 0.273 |
| XComposer-VL-2 [22] | 0.045 | 0.662 | 0.672 | 0.648 | 0.067 | 0.059 |
| Co-Instruct [46] | 0.672 | 0.426 | 0.391 | 0.695 | 0.718 | 0.849 |
| **Compare2Score** | **0.849** | **0.720** | **0.870** | **0.861** | **0.788** | **0.858** |

Table 5: SRCC results for **Compare2Score** using anchor images from KonIQ-10k [7], KADID-10k [26], and AGIQA-3K [61].

| Dataset | KonIQ-10k [7] | KADID-10k [26] | AGIQA-3K [61] |
|---|---|---|---|
| LIVE [53] | 0.972 | 0.968 | **0.975** |
| CSIQ [24] | **0.950** | 0.947 | 0.946 |
| KADID-10k [26] | 0.952 | **0.957** | 0.944 |
| BID [25] | **0.919** | 0.914 | 0.916 |
| CLIVE [54] | 0.914 | 0.912 | **0.915** |
| KonIQ-10k [7] | **0.939** | 0.931 | 0.929 |

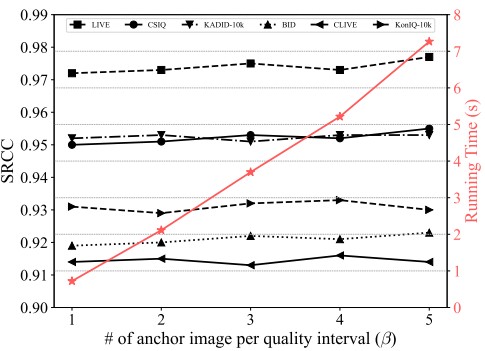

Figure 4: Comparisons of SRCC results and running time with different numbers of anchor image per quality interval ($\beta$).

**Performance of Prediction Accuracy.** We further compare the prediction accuracy of paired comparison results of **Compare2Score** to five open-source LMMs [19, 21, 20, 22, 46]. As shown in Table 4, **Compare2Score** significantly surpasses these advanced open-source LMMs, providing high accuracy of quantitative outputs. Notably, Co-Instruct achieves competitive accuracy across the six IQA datasets, benefiting from a specialized visual quality comparison training corpus. The models (*e.g.*, IDEFICS2, LLaVA-1.5, mPLUG-Owl2, and XComposer-VL-2) rely on high-level instruction-tuning datasets showing poor performance in terms of prediction accuracy, suggesting an inferior quality comparison capability of these LMMs.

## 4.3 Ablation Studies

**Impact of the Source of Anchor Images.** Although KonIQ-10k [7] serves as the default source for anchor images, we demonstrate the robustness of our results across diverse sources of anchor images. Utilizing the same anchor image selection strategy outlined in Eqn. (4), we selected anchor images from three distinct datasets: KADID-10k [26], featuring synthetic distortions; KonIQ-10k [7], with realistic distortions; and AGIQA-3K [61], containing generative distortions. As shown in Table 5, **Compare2Score** consistently shows superior performance across all IQA datasets, showing remarkable robustness to the varying types of distortions of the anchor images. The anchor images selected from the three datasets are shown in Appendix A.5.

**Impact of the Anchor Selection Methods** To evaluate the efficacy of the proposed anchor image selection method (referring to Eqn. (4)), We compare the proposed minimum variance anchor image selection method to the maximum variance and random selection methods. The results are shown in Table 6, from which we can observe that **Compare2Score** achieves the best result among all the testing IQA datasets. This improvement indicates that selecting anchor images with low variability in human ratings is crucial, as high variability tends to introduce noise and biases, compromising the effectiveness of LMMs in performance evaluations. In addition. the other anchor selection methods also demonstrate competitive performance compared with the state-of-the-art NR-IQA model in Table 1, which further verifies the effectiveness of the proposed **Compare2Score** framework.

Table 6: SRCC results of the different anchor selection schemes on KonIQ-10k [7] dataset.

| Method | LIVE [53] | CSIQ [24] | KADID-10k [26] | BID [25] | CLIVE [54] | KonIQ-10k [7] |
|---|---|---|---|---|---|---|
| Random Selection | 0.954 | 0.939 | 0.944 | 0.881 | 0.890 | 0.915 |
| Maximum Variance | 0.958 | 0.940 | 0.926 | 0.885 | 0.879 | 0.919 |
| **Compare2Score** | **0.972** | **0.950** | **0.952** | **0.919** | **0.914** | **0.931** |

**Impact of the Number of Anchor images.** The number of anchor images within each quality interval ($\beta$ in Eqn. (4)) crucially affects the efficacy of **Compare2Score**. We systematically explore the influence of $\beta$ and present the SRCC results and running times in Fig. 4. Notably, all experiments were carried out on the same testing platform equipped with an NVIDIA RTX3090 GPU. The results illustrated in Fig. 4 indicate that increasing the number of anchor images does not enhance performance across the evaluated IQA datasets. $\beta = 1$ suffices for achieving promising and state-of-the-art performance. In addition, it is expected that the average running time (red line in Fig. 4) linearly increases as $\beta$ becomes large. As a result, $\beta = 1$ has been set as the optimal configuration to achieve a practical balance between model efficiency and computational expense.

## 5  Conclusion and Discussion

In this paper, we introduced **Compare2Score**, a novel NR-IQA model utilizing LMM to bridge the gap between discrete comparative levels and continuous quality scores. By using the robust capabilities of LMMs to interpret and integrate complex textual and visual inputs, our model excels in translating scaled-up comparative instructions into reliable, human-like quality assessments. We propose an innovative soft comparison method that effectively and efficiently converts discrete textual responses to continuous quality scores. Extensive validation on standard IQA datasets demonstrates that the proposed model significantly outperforms existing NR-IQA models across different distortion scenarios. Moreover, our probability-matrix-based improves both our model and general-purpose LMMs, showcasing the broad applicability and intrinsic effectiveness of our methods.

**Limitation.** Despite promising, our study has several limitations that highlight areas for future research. While the soft comparison method is effective, it involves computational complexities that may not scale linearly with the increase in the number of images and comparisons. In addition, though LMM provides advanced capabilities in generating human-like quality assessments, the interpretability of this model remains limited. As such, exploring more efficient algorithms and enhancing the interpretability of the LMM is crucial for broader acceptance and trustworthiness in critical applications.

## Acknowledgments

This work was supported in part by the National Key Research and Development Program of China under Grant 2023YFE0210700, the Shenzhen Science and Technology Program under Project JCYJ20220530140816037, the Hong Kong Research Grants Council General Research Fund 11203220, the Innovation and Technology Fund Project GHP/044/21SZ, the Singapore Ministry of Education Tier-2 Research Grant MOE-T2EP20123-0006, and the National Natural Science Foundation of China under Grant 62401214 and 62441203.

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

# A  Appendix

## A.1  Broader Impacts

The social impact of our research on an LMM-based NR-IQA model is substantial. By integrating diverse IQA datasets more effectively, this model promotes enhanced consistency and reliability in image quality evaluations across different real-world applications. Particularly in fields such as image processing, computer vision, and computer graphics, the ability to accurately assess image quality is paramount. As these sectors increasingly rely on visual content, ensuring high-quality, reliable assessments directly influences consumer satisfaction and operational efficiency. Moreover, by advancing the capabilities of LMMs in interpreting and integrating complex, varied data sources, our approach not only improves the precision of image quality scoring systems but also contributes to the broader adoption of responsible AI technologies. This innovation holds promise for setting new benchmarks in the automation of quality control, fostering trust and dependability in digital media and related technologies, ultimately benefiting society at large.

## A.2  More Details on IQA Datasets

The details of the utilized IQA datasets are described as follows.

- **LIVE [53]:** The LIVE database contains 29 reference images and 779 distorted images with the 5 distortions, including JPEG compression, JPEG2000 compression, additive white Gaussian noise, Gaussian blur, and fast-fading transmission distortion. The single-stimulus continuous quality rating is used to collect human opinions. The difference MOSs (DMOS) spread from 0 to 100.

- **CSIQ [24]:** The CSIQ database contains 30 reference images and 866 distorted images with 6 distortions of JPEG compression, JPEG2000 compression, Gaussian blur, Gaussian white noise, Gaussian pink noise, and contrast change. The subjective testing method is the single-stimulus absolute category rating. The DMOSs span from 0 to 1.

- **TID2013 [24]:** The TID2013 database contains 25 reference images and 3,000 distorted images with 24 distortions such as noise (*e.g.*, additive Gaussian, masked, high frequency noise), compression (*e.g.*, JPEG, JPEG2000 compression), contrast change, etc. Paired comparison is the subjective user study methodology. The MOSs spread from 0 to 9.

- **KADID-10k [26]:** The KADID-10k database contains 81 reference images and 10,125 distorted images by adding 25 distortion types with 5 distortion levels such as blur, color distortions, noise, spatial distortions, etc. The subjective testing method is the double-stimulus continuous quality rating by crowdsourcing. The DMOSs range from 1 to 5.

- **BID [25]:** The BID database collects a total of 586 realistic distortion images with profession digital single-lens reflex cameras. The single-stimulus continuous quality rating is applied to collect human opinions. The MOSs range from 0 to 5.

- **CLIVE [54]:** The LIVE Challenge (denoted as CLIVE) database contains 1,162 images with realistic distortions captured by multiple mobile devices. The subjective experiment methodology is the single-stimulus method with crowdsourcing. The MOSs range from 0 to 100.

- **KonIQ-10k [7]:** The KonIQ-10k database consists of 10,073 images with aboundent realistic distortions. Those are selected from the YFCC100M database [64]. Single-stimulus absolute category rating is the method of subjective testing. The MOSs range from 1 to 5.

- **SPAQ [60]:** The SPAQ database consists of 11,125 in-the-wild pictures taken by 66 smartphones. Each picture is annotated with quality, attributes, and scene categories using the single-stimulus methodology. The MOSs range from 0 to 100.

- **AGIQA-3K [61]:** The AGIQA-3K consists of 2,982 AI-generated images derived from 6 advanced text-to-image generation models, which includes AttnGAN [65], DALLE2 [66], GLIDE [67], Midjourney [68], Stable Diffusion [69], and Stable Diffusion XL [70]. The single-stimulus continuous quality rating is used to collect human opinions. The MOSs range from 0 to 5.

Table 7: Overview of the baseline open-sourced LMMs compared with the proposed **Compare2Score**. MLP stands for the multilayer perceptron. MAM is the modality-adaptive module.

| Model | Visual Model | Visual-Language Alignment | Language Model |
|---|---|---|---|
| IDEFICS2 [19] | CLIP-ViT-Large/14 | MLP | Mistral-7B |
| LLaVA-1.5 [23] | CLIP-ViT-Large/14 | MLP | Vicuna-7B |
| mPLUG-Owl2 [20] | CLIP-ViT-Large/14 | MAM | LLaMA-7B |
| XComposer-VL-2 [22] | CLIP-ViT-Large/14 | Perceive Sampler | InternLM2-7B |
| Co-Instruct [46] | CLIP-ViT-Large/14 | MAM | LLaMA-7B |
| **Compare2Score** | CLIP-ViT-Large/14 | MAM | LLaMA-7B |

## A.3   More Details on Competing LMMs

We evaluated the image quality prediction performance of the proposed **Compare2Score** against five LMMs: IDEFICS2 [19], LLaVA-1.5 [21], mPLUG-Owl2 [20], XComposer-VL-2 [22], and Co-Instruct [46]. Typically, LMMs consist of three core components: a modality encoder, a language model, and a modality interface for cross-modal interactions. Detailed information on these models is provided below and summarized in Table 7:

- **IDEFICS2 [19]:** The IDEFICS2 model utilizes a CLIP-based vision encoder and Mistral-7B [71] as the language model, with a visual-text alignment module that facilitates cross-modal interaction, enabling it to handle a variety of multimodal tasks such as image captioning and visual question answering, guided by instruction-tuning on diverse multimodal datasets.

- **LLaVA-1.5 [21]:** The LLaVA-1.5 model incorporates the pretrained CLIP-ViT-L14 [55] as the vision encoder, Vicuna-7B [72] as the LLM and an multilayer perceptron (MLP) as a visual-language connector. LLaVA-1.5 employs the single response formatting prompt in instruction-tuning as formatting control in order to regulate the answering layout.

- **mPLUG-Owl2 [20]:** The mPLUG-Owl2 model integrates the CLIP-ViT-L14 [55] for visual input and LLaMA2-7B [51] as the LLM, with a visual abstractor serving as the cross-modal interface. It emphasizes semantic consistency by decoupling visual-language representations into a shared semantic space, trained on multimodal instructions and image-text pairs.

- **XComposer-VL-2 [22]:** The Intern-XComposer-VL-2 (denoted as XComposer-VL-2) model utilizes the pretrained CLIP-ViT-L14 [55] as the vision encoder, the InternLM-2 [73] as the LLM, and a perceive sampler to connect modalities.

- **Co-Instruct [46]:** The Co-Instruct is built on the mPLUG-Owl2 framework [20], tailored for visual quality comparison using a specialized visual instruction dataset. It employs an image-text interleaved format to enhance the fidelity of information integration, making it highly effective for IQA tasks involving multiple images.

## A.4   Inference Latency

The training process utilizes advanced models like mPLUG-Owl2 and requires substantial computational resources. It takes approximately 20 hours of training on $180,000$ image pairs with a batch size of $64$ across all datasets, spans two epochs, and demands seven NVIDIA A40 GPUs. During inference, a single NVIDIA RTX3090 GPU is sufficient for executing the soft comparison, which is more cost-efficient compared to the training phase. We compute the inference latency of our method with different batch sizes. All experiments are conducted on the same device with a single RTX3090 GPU. The results are shown in Table 8, from which we can observe that the latency of the inference process increases with the batch size. For instance, the latency for a batch size of 1 is $0.931$ seconds, while for a batch size of $64$, it is $45.263$ seconds. This demonstrates the scalability of our model during inference, allowing for flexible adaptation based on the available computational resources and required processing speed.

Table 8: Inference latency of the **Compare2Score** with different batch sizes on RTX3090.

| Batch size | 1 | 2 | 4 | 8 | 16 | 32 | 64 |
|---|---|---|---|---|---|---|---|
| Latency (s) | 0.931 | 1.739 | 3.644 | 6.581 | 11.125 | 23.365 | 45.263 |

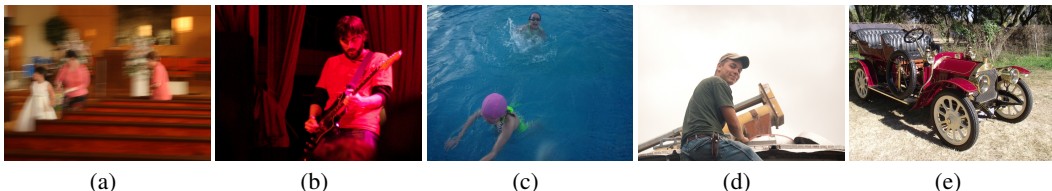

|     |     |     |     |     |
|-----|-----|-----|-----|-----|
| (a) | (b) | (c) | (d) | (e) |

Figure 5: Illustration of the five anchor images selected from KonIQ-10k [7]. (a) MOS = 1.09, $\sigma = 0.29$; (b) MOS = 2.02, $\sigma = 0.39$; (c) MOS = 2.96, $\sigma = 0.38$; (d) MOS = 3.21, $\sigma = 0.41$; (e) MOS = 4.01, $\sigma = 0.34$.

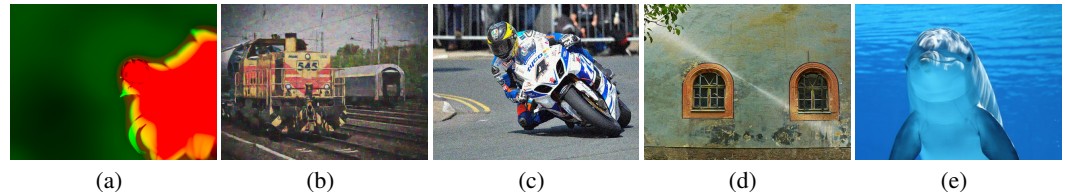

|     |     |     |     |     |
|-----|-----|-----|-----|-----|
| (a) | (b) | (c) | (d) | (e) |

Figure 6: Illustration of the five anchor images selected from KADID-10k [26]. (a) MOS = 1.00, $\sigma = 0.00$; (b) MOS = 1.80, $\sigma = 0.40$; (c) MOS = 2.84, $\sigma = 0.51$; (d) MOS = 3.97, $\sigma = 0.41$; (e) MOS = 4.90, $\sigma = 0.30$.

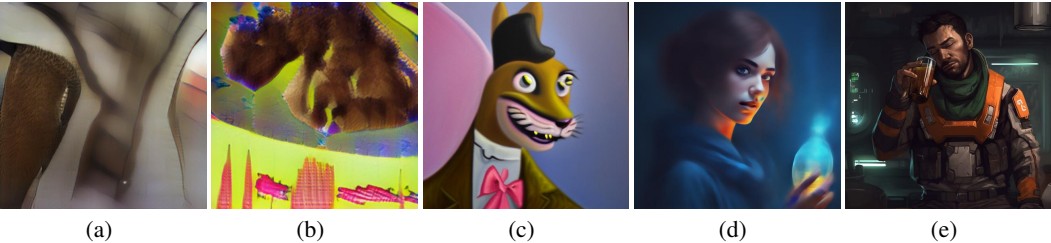

|     |     |     |     |     |
|-----|-----|-----|-----|-----|
| (a) | (b) | (c) | (d) | (e) |

Figure 7: Illustration of the five anchor images selected from AGIQA-3K [61]. (a) MOS = 0.73, $\sigma = 0.10$; (b) MOS = 0.95, $\sigma = 0.12$; (c) MOS = 2.27, $\sigma = 0.14$; (d) MOS = 3.41, $\sigma = 0.16$; (e) MOS = 3.96, $\sigma = 0.17$.

## A.5 Illustrations of Anchor Images

Herein, we present the selected anchor images from KonIQ-10k [7], KADID-10k [26], and AGIQA-3K [61] in Figs. 5, 6, and 7, respectively. We can observe the selected images cover a wide range of visual quality, and the contents of the images are diverse. This robustness of the selection of anchor images effectively supports the model's ability to generalize across different types of visual distortions, enhancing its applicability in real-world IQA scenarios.

