# OpenReview forum: "Adaptive Image Quality Assessment via Teaching Large Multimodal Model to Compare"
_NeurIPS.cc/2024/Conference — NeurIPS 2024 spotlight_

### Official Review · Reviewer_nq4X · 2024-06-14

**Soundness:** 3
**Presentation:** 4
**Contribution:** 3
**Rating:** 7
**Confidence:** 4

**Summary:**

The paper introduces an LMM-based no-reference IQA framework that can generate qualitative comparisons between images and translate those discrete comparisons into continuous quality scores. During training, it generates comparative instructions by comparing image pairs within the same IQA dataset, allowing flexible integration of multiple datasets. At inference, it uses a soft comparison by computing the likelihood of the test image being preferred over multiple anchor images, followed by MAP estimation to obtain the final quality score. Extensive experiments validate the state-of-the-art performance across synthetic and realistic distortions.

**Strengths:**

1.	Well-justified motivation for using relative comparisons instead of absolute ratings to combine IQA datasets.
2.	The "soft comparison" inference strategy is innovative and effective for scoring images for relative quality comparison.
3.	State-of-the-art performance on benchmark IQA datasets under various test condition.
4.	The paper is well-written and easy to follow.

**Weaknesses:**

The authors do not provide an in-depth analysis of the model's generalization capabilities on unseen distortions and datasets.

**Questions:**

1.	In Table 3, the results on KADID-10k and KonIQ-10k should also be given.
2.	The authors use the MAP estimation to compute the quality scores. How about using other quality aggregation methods in [49].
3.	The impact of the anchor image selection strategy could be further explored, such as investigating different methods for anchor image selection.
4. Typo: KonIQ-10K -> KonIQ-10k.

**Limitations:**

While the authors have identified some limitations in the conclusion section, a more comprehensive discussion on strategies to address the limitations would strengthen the paper and provide valuable insights for future research.

---

> ### Author Rebuttal · Authors · 2024-08-05
>
> Thanks for recognizing the merits and strengths of our paper. Point-to-point responses are given as follows.
>
> **Q1. The generalization capabilities.**
>
> **A1:** To assess the generalization capability of Compare2Score to unseen distortions and datasets, we conduct the cross-distortion experiments with three unseen IQA datasets: TID2013, SPAQ, and AGIQA-3K in Sec. 4.2 of the manuscript. TID2013 contains 24 distortion types, most of which are different from distortions in the training datasets. SPAQ consists of 11,125 images captured by 66 smartphones, undergoing abundant realistic distortions. The images in AGIQA-3K are generated by six advanced text-to-image generative models, which cast significant challenges to the NR-IQA models. The results are summarized in Table 2 of the manuscript, from which we can observe that the proposed method demonstrates the strongest generalization capability across synthetic, realistic, and generative distortions.
>
> **Q2. More results on KADID-10k and KonIQ-10k.**
>
> **A2:** We have included the results for the KADID-10k and KonIQ-10k datasets in Table 1, which comprehensively includes the SRCC results for all six IQA datasets. It compares the performance of different methods using both probability and count matrices. This complete set of results provides a clearer understanding of the model's performance across various datasets and confirms the effectiveness of the Compare2Score framework in diverse scenarios. We will include these results in our revised manuscript.
>
> TABLE 1: SRCC results of probability matrix and count matrix on  KADID-10k and KonIQ-10k
>
> | Method               | Matrix | KADID-10k  | KonIQ-10k  |
> |----------------------|--------|---------------|---------------|
> | IDEFICS          | Count  | 0.204         | 0.245         |
> |                      | Prob.  | 0.217         | 0.301         |
> | LLaVA-1.5       | Count  | 0.151         | 0.205         |
> |                      | Prob.  | 0.217         | 0.297         |
> | mPLUG-Owl2       | Count  | 0.183         | 0.295         |
> |                      | Prob.  | 0.245         | 0.343         |
> | XComposer-VL-2   | Count  | 0.140         | 0.145         |
> |                      | Prob.  | 0.223         | 0.243         |
> | Co-Instruct      | Count  | 0.496         | 0.708         |
> |                      | Prob.  | 0.745         | 0.832         |
> | **Compare2Score**    | Count  | 0.921         | 0.889         |
> |                      | Prob.  | **0.952**     | **0.931**     |
>
> **Q3. Other quality aggregation methods in [49].**
>
> **A3:**  We conduct experiments comparing the maximum likelihood estimation (MLE) [49] and the Perron method with our maximum a posterior (MAP) estimation approach. The results, presented in Table 2, indicate that while MLE and the Perron method offer competitive performance, MAP estimation consistently achieves better SRCC and PLCC values across various datasets. This suggests that MAP estimation is well-suited for our soft comparison methodology.
>
> TABLE 2: Performance comparison in terms of median SRCC and PLCC on six IQA datasets.
>
> | Method            | LIVE   |          | CSIQ   |          | KADID-10k  |          | BID   |          | CLIVE   |          | KonIQ-10k  |          |
> |-------------------|-----------|----------|-----------|----------|---------------|----------|----------|----------|------------|----------|---------------|----------|
> |                   | SRCC      | PLCC     | SRCC      | PLCC     | SRCC          | PLCC     | SRCC     | PLCC     | SRCC       | PLCC     | SRCC          | PLCC     |
> | MLE               | 0.969     | 0.962    | **0.952** | 0.941    | 0.948         | 0.944    | 0.912    | 0.935    | 0.913      | 0.925    | 0.925         | 0.938    |
> | Perron            | 0.956     | 0.960    | 0.941     | **0.944**    | 0.943         | **0.945**| 0.911    | 0.938    | 0.911      | 0.920    | 0.915         | 0.924    |
> | **Compare2Score** | **0.972** | **0.969**| 0.950     |  0.943 | **0.952**     | 0.939    | **0.919**| **0.939**| **0.914**  | **0.928**| **0.931**     | **0.939**|
>
> **Q4. The anchor image selection methods.**
>
> **A4:** We compare the proposed minimum variance anchor image selection method to the maximum variance and random selection methods. The results are shown in Table 3, where we can observe that the minimum variance value achieves the best result among all the testing IQA datasets. As such, ensuring low variance in the selected anchor images is crucial, as it minimizes noise and biases, leading to more accurate and robust quality assessments in the model.
>
> TABLE 3: SRCC results of different anchor selection schemes.
>
> | Method                        | LIVE  | CSIQ  | KADID-10k  | BID  | CLIVE | KonIQ-10k |
> |-------------------------------|----------|----------|---------------|---------|-----------|---------------|
> | Random Selection              | 0.954    | 0.939    | 0.944         | 0.881   | 0.890     | 0.915         |
> | Maximum Variance                   | 0.958    | 0.940    | 0.926         | 0.885   | 0.879     | 0.919         |
> | **Minimum Variance** | **0.972** | **0.950** | **0.952**     | **0.919** | **0.914**   | **0.931**    |
>
>
> **Q5. Typo: KonIQ-10K -> KonIQ-10k.**
>
> **A5:**  We have revised the typo, and careful proofreading has been conducted to improve the quality of our paper further.
>
> **Q6. Strategies to address the limitations in the future.**
>
> **A6:** To address the computational complexities of the soft comparison method, we plan to develop more efficient algorithms, for example parallel processing technique, to ensure scalability with increasing images. Additionally, improving the interpretability of the LMM is crucial. We aim to incorporate explainable AI techniques, such as attention visualization and model-agnostic interpretability tools, to enhance understanding of the LMM's decision-making process. These strategies will be pivotal for broader acceptance and trustworthiness in critical applications.

---

### Official Review · Reviewer_WZ3r · 2024-07-01

**Soundness:** 4
**Presentation:** 4
**Contribution:** 3
**Rating:** 7
**Confidence:** 5

**Summary:**

The paper presents a framework that trains an LMM as a visual quality comparator using relative image comparisons, and converts the discrete comparison outputs to continuous quality scores via a soft comparison method. It generates paired image comparisons from existing IQA datasets to train the LMM, and uses a probability matrix during inference to determine quality scores. Experiments demonstrate the proposed model outperforms prior NR-IQA methods on several synthetic and realistic distortions datasets.

**Strengths:**

1.The proposed training strategy using comparative instructions is well-designed and addresses the challenge of combining different IQA datasets with varying perceptual scales.

2.The soft comparison method provides a more nuanced and informative approach to inference compared to traditional binary choices.

3.The paper demonstrates the effectiveness of Compare2Score through extensive experiments on nine IQA datasets, showing significant improvements over state-of-the-art models.

**Weaknesses:**

1.In Tables 3 and 4, the performance on IDEFICS2 should be included, which is the latest version of IDEFICS family.

2.In Fig. 4, more details on computing the running time should be given, such as the device and input images.

3.The authors do not provide any indication of plans to open-source the code for Compare2Score, which may limit the replicability of the proposed approach by the research community.

4.The title should be revised from "adaptive image quality assessment via teaching large multimodal models to compare" to "adaptive image quality assessment via teaching a large multimodal model to compare", as the paper only utilizes one large multimodal model, mPLUG-Owl-2, rather than multiple models.

**Questions:**

Please address the comments in the weaknesses section.

**Limitations:**

The authors have thoughtfully discussed the paper's limitations in the conclusion, but one aspect that could be further discussed is the impact of anchor image selection on the model's performance and potential biases.

---

> ### Author Rebuttal · Authors · 2024-08-05
>
> Thanks for recognizing the merits and strengths of our paper. Point-to-point responses to specific comments are given as follows.
>
> **Q1. Including the IDEFICS2 in Tables 3 and 4**
>
> **A1:** We have included the performance of IDEFICS2, the latest version of the IDEFICS family. As shown in Tables 1 and 2 of the reponse letter IDEFICS2 demonstrates improved performance over its predecessor IDEFICS, but our proposed method still outperforms both versions in terms of SRCC and prediction accuracy across various datasets. This inclusion provides a more complete and current comparison, further validating the robustness and effectiveness of Compare2Score. We will update these results in our revised manuscript.
>
> TABLE 1: SRCC results of probability matrix and count matrix on six IQA datasets
>
> | Method            | Matrix | LIVE  | CSIQ  | KADID-10k  | BID  | CLIVE  | KonIQ-10k  |
> |-------------------|--------|----------|----------|---------------|---------|-----------|---------------|
> | IDEFICS       | Count  | 0.157    | 0.008    | 0.204         | 0.015   | 0.206     | 0.245         |
> |                   | Prob.  | 0.363    | 0.044    | 0.217         | 0.227   | 0.385     | 0.301         |
> | IDEFICS2      | Count  | 0.354    | 0.208    | 0.198         | 0.292   | 0.360     | 0.481         |
> |                   | Prob.  | 0.465    | 0.567    | 0.389         | 0.436   | 0.392     | 0.517         |
> | **Compare2Score** | Count  | 0.888    | 0.875    | 0.921         | 0.778   | 0.816     | 0.889         |
> |                   | Prob.  | **0.974**| **0.942**| **0.952**     | **0.921**| **0.934** | **0.931**     |
>
> TABLE 2: Performance comparison in terms of prediction accuracy on six IQA datasets
>
> | Method            | LIVE   | CSIQ  | KADID-10k  | BID  | CLIVE  | KonIQ-10k  |
> |-------------------|-----------|----------|---------------|---------|-----------|---------------|
> | IDEFICS       | 0.125     | 0.669    | 0.500         | 0.523   | 0.146     | 0.727         |
> | IDEFICS2      | 0.453     | 0.546    | 0.521         | 0.566   | 0.407     | 0.687         |
> | **Compare2Score** | **0.849** | **0.720**| **0.870**     | **0.861**| **0.788** | **0.858**     |
>
> **Q2. Details on computing the running time should be given, such as the device and input images.**
>
> **A2:** All running time measurements were conducted using a single NVIDIA RTX3090 GPU. The input images used for these computations were resized to a resolution of 448x448 pixels. The running time includes both the LMM inference and the soft comparison stages of the process.  We will include these details in the final version of our paper to ensure clarity and comprehensiveness.
>
> **Q3. Open-source the code.**
>
> **A3:**  We have included the source code in the supplementary materials. Upon acceptance of the paper, we will publish the code, along with detailed documentation and example scripts, on a public GitHub repository.
>
> **Q4. Regarding the title.**
>
> **A4:** We agree that the title should accurately reflect the content of the paper. Therefore, we will revise the title to "Adaptive Image Quality Assessment via Teaching a Large Multimodal Model to Compare," as the paper focuses on utilizing a single large multimodal model, mPLUG-Owl-2. This change will provide better clarity and accurately represent the scope of our work.
>
> **Q5. Discussing the impact of anchor image selection on the model's performance and potential biases.**
>
> **A5:** We compare the proposed minimum variance anchor image selection method to the maximum variance and random selection methods. The results are shown in Table 3, where we can observe that the minimum variance value achieves the best result among all the testing IQA datasets. As such, ensuring low variance in the selected anchor images is crucial, as it minimizes noise and biases, leading to more accurate and robust quality assessments in the model.
>
> TABLE 3: SRCC results of different anchor selection schemes
>
> | Method                        | LIVE  | CSIQ  | KADID-10k  | BID  | CLIVE | KonIQ-10k |
> |-------------------------------|----------|----------|---------------|---------|-----------|---------------|
> | Random Selection              | 0.954    | 0.939    | 0.944         | 0.881   | 0.890     | 0.915         |
> | Maximum Variance                   | 0.958    | 0.940    | 0.926         | 0.885   | 0.879     | 0.919         |
> | **Minimum Variance** | **0.972** | **0.950** | **0.952**     | **0.919** | **0.914**   | **0.931**    |

---

### Official Review · Reviewer_WPaz · 2024-07-11

**Soundness:** 3
**Presentation:** 3
**Contribution:** 3
**Rating:** 7
**Confidence:** 4

**Summary:**

This paper introduces Compare2Score, a novel NR-IQA model that harnesses the robust capabilities of LMM to interpret and integrate complex textual and visual inputs. The model is trained using a relative quality comparison strategy. Additionally, the authors propose a soft comparison approach that transforms discrete textual responses into continuous quality scores. Experiments conducted on nine IQA datasets validate its effectiveness.

**Strengths:**

1. The motivation behind the method is clearly articulated. By integrating pairwise comparisons into the foundational model, the approach effectively tackles the data challenge problem prevalent in IQA.
2. This paper is easy to follow, providing a clear and coherent explanation of the proposed visual quality comparator and inference conversion strategy. The logical soundness of the method is well-articulated, ensuring transparency in its approach and fostering understanding among readers.
3. The method achieves promising results across nine IQA datasets, demonstrating its efficacy and robust performance in various evaluation scenarios.
4. Good reproducibility, code provided.

**Weaknesses:**

1. How is the standard deviation determined when constructing image pairs? How variations in standard deviation can affect the pairing process and subsequent quality scores? The impact of this factor should be discussed.
2. Cost analyses are required.
3. Ablation studies on the network structure are insufficient.

**Questions:**

Please refer to the weaknesses section.

**Limitations:**

Limitations are discussed and resolved.

---

> ### Author Rebuttal · Authors · 2024-08-05
>
> Thanks for recognizing the merits and strengths of our paper. Point-to-point responses to specific comments are given as follows.
>
> **Q1. How is the standard deviation determined when constructing image pairs? How do variations in standard deviation affect the pairing process and subsequent quality scores?**
>
> **A1:** The standard deviation (STD) in the Compare2Score framework is determined based on the variability of mean opinion scores (MOS) from subjective testing within each dataset.
>
> **Regarding the Construction of Image Pairs:** Image pairs are categorized into five comparative levels using the empirical rule. Specifically, the quality difference $ q^{(ij)} = q^{(i)} - q^{(j)} $ of an image pair is used to determine the comparative level, which is assessed by the corresponding STD $ \sigma^{(ij)} = \sqrt{(\sigma^{(i)})^2 + (\sigma^{(j)})^2}$. As summarized in Eqn. (2) of the manuscript, significance thresholds at $\pm\sigma^{(ij)}$ and $\pm2\sigma^{(ij)}$ effectively categorize quality differences into various comparative levels.
>
> **Regarding the Impact on Quality Scores:** Since we use the variance to select the achor images, it also significantly impact quality scores. STD represents the confidence in the quality scores of the images. Larger STD indicates less confidence in the MOS, leading to potentially inconsistent quality assessments. Conversely, Smaller STD indicates higher confidence in the MOS, resulting in more consistent quality scores. As such , we compared the minimum variance to the maximum variance and random selection of each quality interval. The results, shown in Table 1 of the response letter, demonstrate that the minimum variance achieves the best result across all tested IQA datasets. Thus, ensuring low variability in the selected anchor images is crucial, as it minimizes noise and biases, leading to more accurate and robust quality assessments.
>
> TABLE 1: SRCC results of different anchor selection schemes
>
> | Method                        | LIVE  | CSIQ  | KADID-10k  | BID  | CLIVE | KonIQ-10k |
> |-------------------------------|----------|----------|---------------|---------|-----------|---------------|
> | Random Selection              | 0.954    | 0.939    | 0.944         | 0.881   | 0.890     | 0.915         |
> | Maximum Variance                   | 0.958    | 0.940    | 0.926         | 0.885   | 0.879     | 0.919         |
> | **Minimum Variance** | **0.972** | **0.950** | **0.952**     | **0.919** | **0.914**   | **0.931**    |
>
>
> **Q2. Cost analyses are required.**
>
> **A2:** The training process utilizes advanced models like mPLUG-Owl2 and requires substantial computational resources. It takes approximately 20 hours of training on 180,000 image pairs with a batch size of 64 across all datasets, spans two epochs, and demands seven NVIDIA A40 GPUs. During inference, a single NVIDIA RTX3090 GPU is sufficient for executing the soft comparison, which is more cost-efficient compared to the training phase. We compute the inference latency of our method with different batch sizes. All experiments are conducted on the same device with a single RTX3090 GPU. The results are shown in Table 2 of the response letter, from which we can observe that the latency of the inference process increases with the batch size. For instance, the latency for a batch size of 1 is 0.931 seconds, while for a batch size of 64, it is 45.263 seconds. This demonstrates the scalability of our model during inference, allowing for flexible adaptation based on the available computational resources and required processing speed.
>
> TABLE 2: Inference latency of the Compare2Score with different batch sizes on RTX3090
>
> | Batch size | 1     | 2     | 4     | 8     | 16    | 32     | 64      |
> |------------|-------|-------|-------|-------|-------|--------|---------|
> | Latency (s)| 0.931 | 1.739 | 3.644 | 6.581 | 11.125 | 23.365 | 45.263  |
>
> **Q3. Ablation studies on the network structure are insufficient.**
>
> **A3:** Thank you for your insightful comment. Compare2Score utilizes the advanced mPLUG-Owl2 model for its architecture, leveraging a pre-trained CLIP-ViT-L14 as the vision encoder and LLaMA2-7B as the LLM decoder. To verify the network structure of the model, we have included comparisons with the Freeze Vision and Freeze LLM variants of the network structure in Table 3 as follows. Our findings indicate that both variants perform worse than the default setting, with the Freeze LLM variant showing a particularly significant drop in performance. This discrepancy is likely due to the reliance on the LLM decoder to learn new capabilities for comparing a pair of images, while the visual encoder only embeds vision information independently. These results further validate the effectiveness of each component of the network structure.
>
> TABLE 3: Ablation Studies on the network structure in terms of SRCC
>
> |                       | Vision | LLM | LIVE  | CSIQ  | KADID-10k | BID   | CLIVE | KonIQ-10k |
> |-----------------------|--------|-----|-------|-------|-----------|-------|-------|-----------|
> | mPLUG-Owl2            | ✗      | ✗   | 0.449 | 0.129 | 0.245     | 0.551 | 0.335 | 0.343     |
> | Freeze Vision         | ✗      | ✓   | 0.875 | 0.848 | 0.875     | 0.817 | 0.854 | 0.862     |
> | Freeze LLM            | ✓      | ✗   | 0.775 | 0.694 | 0.775     | 0.673 | 0.789 | 0.738     |
> | **Compare2Score** | ✓      | ✓   | **0.972** | **0.950** | **0.952**     | **0.919** | **0.914** | **0.931**     |

---

> > ### Comment · Reviewer_WPaz · 2024-08-12
> >
> > Thank you for the clarifications. It has addressed most of my concerns, so I will be raising my original score.

---

> > > ### Author Response · Authors · 2024-08-14
> > >
> > > Dear Reviewer WPaz,
> > >
> > > We are glad to hear that your concerns have been addressed. Thanks for keeping your positive view of our paper.
> > >
> > > Best regards,
> > >
> > > Authors of Submission #937

---

### Official Review · Reviewer_qViG · 2024-07-12

**Soundness:** 4
**Presentation:** 4
**Contribution:** 3
**Rating:** 7
**Confidence:** 4

**Summary:**

This work presents a method named Compare2Score, which is capable of producing qualitatively comparative responses and effectively translating these discrete comparative levels into a continuous quality score. The method utilizes the predefined anchor images to calculates the likelihood and get the quality score. Extensive experiments verifies the effectiveness of the proposed method.

**Strengths:**

1. The work utilizes anchor images to bridge the gap between discrete rank order and continuous score, which is sound and reasonable.
2. A probability matrix is introduced and computed to infer the quality score with MAP estimation.
3. The motivation and writing are neat.

**Weaknesses:**

1. It is still somewhat unclear how to utilize the anchor images to align the difference scales among datasets. Are these anchor images from the same dataset or not? And how to guarantee the effectiveness when the anchor image and test image are from different distribution.
2. The first contribution [A repurposed training dataset] is not a main contribution, which is a general approach.
3. The method does not perform well on authentic datasets like CLIVE and KonIQ compared with Q-Align. It is weird and please try to explain.

**Questions:**

I have some concerns on the details on aligning different scales, overstated contributions, and wish to see more explanation about the experimental results.

---

> ### Author Rebuttal · Authors · 2024-08-05
>
> Thanks for recognizing the merits of our work and for your insightful suggestions. Point-to-point responses to specific comments are given as follows.
>
>
> **Q1. It is still somewhat unclear how to utilize the anchor images to align the difference scales among datasets. Are these anchor images from the same dataset or not? And how to guarantee the effectiveness when the anchor image and test image are from different distribution.**
>
> **A1:** **Regarding Alignment of Scales**: By comparing the mean opinion scores (MOS) of each image within the same dataset, we convert absolute quality ratings into relative quality comparisons. This approach allows for the flexible integration of multiple IQA datasets with different perceptual quality scales. In addition, anchor images provide a common reference point for comparing test images. During the inference phase, the test image is compared against these anchor images. The model calculates the probability of the test image being preferred over each anchor image, constructing a probability matrix. This matrix is further refined using maximum a posteriori estimation, which facilitates normalize the quality scores by aligning them to the scales defined by the anchor images.
>
>
> **Regarding Selection of Anchor Images**: The anchor images are selected from a specific dataset. As indicated in the paper, the default source for anchor images is KonIQ-10k, which contains realistic distortions. However, as shown in Table 5 of the manuscript, we validate that anchor images can be chosen from other datasets as well, such as KADID-10k (synthetic distortions) or AGIQA-3K (generative distortions).
>
> **Regarding Handling Different Distributions:** The effectiveness of the anchor image selection process is crucial to ensure robust performance even when the test image and anchor images are from different distributions. To achieve this, the paper proposes a thorough selection strategy:
>
> - **Low Variability Selection:** Anchor images are selected based on their minimal variance in MOS, ensuring that these images have consistent quality ratings.
> - **Quality Intervals:** The dataset is divided into multiple quality intervals, and representative anchor images are selected from each interval to cover a wide range of perceptual qualities.
>
> As shown in Table 5 of the manuscript, extensive experiments validate the effectiveness of the anchor images. The results show that even when anchor images and test images come from different distributions (synthetic, realistic, or generative), the Compare2Score model maintains superior performance and generalization capabilities. The robustness of the model is attributed to the high capacity of the LMM-based model and the adaptive soft comparison mechanism, which effectively handles the alignment across diverse distortion scenarios.
>
> **Q2. The first contribution [A repurposed training dataset] is not a main contribution, which is a general approach.**
>
> **A2:** While leveraging comparative instructions and generating datasets by comparing within the same dataset is a general approach in machine learning applications, the specific implementation tailored to IQA and its integration with an LMM is novel. It is also worth noting that the tailored dataset creation facilitates the flexible combination of multiple IQA datasets, addressing the challenge of differing subjective testing methodologies and perceptual scales. This flexibility is crucial for improving the generalization capability of the IQA model across diverse datasets. However, to strengthen the presentation of this contribution, we emphasize the unique aspects and innovations specific to this contribution as follows.
>
> **[A repurposed training dataset.]** We introduce a tailored approach to generate comparative instructions by comparing MOSs within each IQA dataset. This method categorizes image pairs into distinct comparative levels (inferior, worse, similar, better, superior) using the empirical rule, facilitating the flexible integration of diverse IQA datasets. This specific implementation effectively addresses the challenges posed by differing subjective testing methodologies and perceptual scales. It produces a comprehensive training dataset that enables the LMM to handle various distortion scenarios, resulting in a human-like **visual quality comparator**.
>
> **Q3. The method does not perform well on authentic datasets like CLIVE and KonIQ compared with Q-Align. It is weird and please try to explain.**
>
> **A3:** The discrepancy in performance between Compare2Score and Q-Align on authentic datasets like CLIVE and KonIQ-10k may be caused by the following factors. Q-Align's reliance on absolute quality ratings might better capture the nuanced and diverse distortions present in real-world images. In contrast, Compare2Score's method, based on relative comparisons, may face challenges in translating the nuanced quality discrepancy into accurate quality scores for the high variability of authentic distortions. Additionally, differences in the training data distribution, with Q-Align potentially being better fine-tuned for authentic datasets, and the inherent complexity of real-world distortions that might not be fully captured by relative comparisons alone, could contribute to this performance gap.

---

### Decision · Program_Chairs · 2024-09-25

**Decision:**

Accept (spotlight)

**Comment:**

The work proposes to train an LMM as a visual quality assessor using image comparisons and a soft comparison method to convert discrete to continuous quality scores. It employs existing IQA datasets for training and a probability matrix at runtime to produce scores. Its effectiveness is shown on nine NR-IQA benchmarks with synthetic and realistic distortions.

The authors provided satisfactory responses and convinced Reviewer WPaz to increase the score. Thus, for this work there is a consensus on the positive side (4xAccept) from all Reviewers.

After reading the paper and carefully checking the reviews and the authors' responses the ACs agree with the reviews that the present work makes important contributions and is of interest for the community.

The authors are invited to further refine their paper for the camera ready by including (part of) information/details from their responses to the reviewers' comments.